# Factors associated with the occurrence and persistence of subthreshold and full attention-deficit hyperactivity disorder in women: A population-based epidemiological study

Ana Buadze[1], Vladeta Ajdacic-Gross[1]*, Mario Müller[1], Yanhua Xu[1], Erich Seifritz[1], En-Young N. Wagner[2], Roland von Känel[2], Jennifer Glaus[3], Setareh Ranjbar[4], Enrique Castelao[4], Marie-Pierre F. Strippoli[4], Martin Preisig[4], Caroline L. Vandeleur[4]

**1** Department of Adult Psychiatry and Psychotherapy, University Hospital of Psychiatry and University of Zurich, Zurich, Switzerland, **2** Department of Consultation-Liaison Psychiatry and Psychosomatic Medicine, University Hospital Zurich, University of Zurich, Zurich, Switzerland, **3** Department of Psychiatry, Service of Child and Adolescent Psychiatry, Lausanne University Hospital and University of Lausanne, Lausanne, Switzerland, **4** Department of Psychiatry, Psychiatric Epidemiology and Psychopathology Research Center, Lausanne University Hospital and University of Lausanne, Prilly, Switzerland

* vladeta.ajdacic-gross@uzh.ch

## Abstract

### Background

Attention-deficit hyperactivity disorder (ADHD) is one of the most frequent neurodevelopmental disorders in youth, and persists into adulthood in the majority of cases. Our objective was to examine which factors are associated with ADHD in women across different disorder levels (subthreshold, full, persistent).

### Methods

Analyses were carried out using data from the CoLaus|PsyCoLaus study (N = 2741, age range 35–88 years). The factors of interest comprised familial aggregation, age of ADHD onset, somatic comorbidity, adverse childhood experiences and trauma, parental bonding, as well as immunological and cardio-metabolic markers. Association analyses were combined with a person-centered analysis approach (latent class/ latent profile analysis (LCA/ LPA)).

### Results

Subthreshold and full ADHD both comprised higher levels of adverse childhood experiences and lower scores of parental bonding. Familial aggregation and earlier age of onset were prominent features in full ADHD, but less relevant regarding persistence. There, increased levels of immunological and cardiometabolic markers played the major role (monocytes, neutrophils, hsCRP, insulin and leptin). In a subgroup perspective, adverse childhood experiences were the most prominent feature of the

**Data availability statement:** Individual-level data cannot be publicly shared by the authors because they contain sensitive, potentially identifiable personal information, and public data sharing was not included in the in-formed consent provided by participants at the start of the CoLaus|PsyCoLaus study (approval by the Ethics Committee of the Canton of Vaud for human research (www.cer-vd.ch; project num-ber PB_2018-00038, reference 239/09)). All relevant aggregated data are available within the paper and its Supporting Information files, where examples of low-level aggregated data are also provided. However, individual-level data from the CoLaus|PsyCoLaus cohort may be accessed by qualified researchers upon reasonable request and following submission of a research application to the research com-mittee of Colaus|PsyCoLaus (see https://www.colaus-psycolaus.ch/professionals/how-to-collaborate), which evaluates requests and grants access in accordance with study regulations.

**Funding:** The CoLaus|PsyCoLaus study (represented by MP) is supported by research grants from GlaxoSmithKline (www.gsk.com), the Faculty of Biology and Medicine of the University of Lausanne (https://www.unil.ch/fbm/en/home/menuinst/faculte.html), and the Swiss National Science Foundation (www.snf.ch) (grants 3200B0–105993, 3200B0-118308, 33CSCO-122661, 33CS30-139468, 33CS30-148401, 33CS30_177535 and 3247730_204523) and the Swiss Personalized Health Network (sphn.ch) (grant 2018DRI01). The funders had no role in study design, data collection and analysis, decision to publish, or preparation of the manuscript.

**Competing interests:** The authors have declared that no competing interests exist.

**Abbreviations:** ADHD: attention deficit hyperactivity disorder, ABIC: sample-size adjusted BIC, ACE: adverse childhood experiences, AIC: Akaike information criterion, AUC: area under the curve, BIC: Bayesian information criterion, BMI: body mass index, BP: blood pressure, DIGS: Diagnostic Interview for Genetic Studies, DSM: Diagnostic and Statistical Manual of Mental Disorders, EEG: electro-encephalography, ECG: electrocardiography, FH-RDC: Family History–Research Diagnostic

"burdened" opposed to the "idiopathic" LCA/LPA subgroup. The "burdened" subgroup showed higher persistence rates than the "idiopathic" subgroup in subthreshold and full ADHD.

## Conclusions

The full picture of subthreshold, full and persistent ADHD in women was characterized by surprisingly heterogeneous association patterns. The heterogeneity does not only rely on severity levels but is primarily associated with additional factors, including familial aggregation in full ADHD or proinflammatory immune and cardiovascular system dysregulation in persistent ADHD.

## Introduction

Attention-deficit hyperactivity disorder (ADHD) is a neuro-developmental disorder that occurs with a worldwide prevalence of approximately 5% in children and often persists into adulthood [1,2]. The initial sex-ratio male:female is 2–3:1 [3,4]. Heredity plays an important role in etiopathogenesis [5,6]. The core symptoms of the disorder are attention problems and hyperactivity/ impulsivity, which can contribute to a significant level of distress depending on the severity and age of affected individuals [7,8]. This, in combination with disorganization, often leads to significant impairment in everyday life. Individuals affected by ADHD have an increased risk for somatic comorbidities such as cardiovascular problems or obesity, as well as for psychiatric disorders [9]. They have a higher risk to develop substance use disorders (odds ratios (OR) up to 8), depression (OR up to 6) and/or anxiety disorders (OR up to 17) [10].

Risk factors, associated factors and mechanisms that are related to ADHD are believed to be poorly understood despite the fact that ADHD has been widely studied and recognized as an important public health concern [8,11]. Epidemiological studies have shown that prenatal and perinatal risk factors, like premature birth, low birth-weight, parental tobacco use, alcohol and drug use during pregnancy and maternal immune activation may affect the development of threshold and subthreshold ADHD [12,13]. Among postnatal risk factors, deprivation [14,15], maltreatment [16] and other childhood adversities [17,18] – also labeled as adverse childhood experiences (ACE) – have been shown to play a prominent role. However, all this apparently represents a small part of the not yet completely clarified multifactorial etiology of ADHD [19,20]. This has led many scholars to believe that environmental factors have a causal impact on ADHD only in combination with genetic factors, or are correlates of the latter [6].

After a long period of neglect, sex-specific differences in ADHD have been receiving more and more attention since the 2000s years, in particular by including findings related to girls and women [4]. When focusing on risk and associated factors, their spectrum seems to be similar for both sexes [21], with few exceptions. With regard to obstetric complications, preterm labor comprised an elevated risk for boys, whereas

Criteria, HbA1c: hemoglobin A1c, hsCRP: high sensitivity C-reactive protein, IBS:irritable bowel syndrome, K-SADS-E: Kiddie-Schedule for Affective Disorders and Schizophrenia – Epidemiologic version, LCA: latent class analysis, LMR-LRT: adjusted Lo-Mendell-Rubin likelihood ratio test, LPA: latent profile analysis, LRT: bootstraped likelihood ratio test, OR: odds ratio(s), PBI: Parental Bonding Instrument, PTSD: post-traumatic stress disorder, SADS-LA: Schedule for Affective Disorders and Schizophrenia – Lifetime and Anxiety disorder version.

oxytocin augmentation reduced the risk for girls [22]. Moreover, girls were shown to be victims of bullying more often than boys [23], to be diagnosed with ADHD significantly later on than boys or to remain untreated more frequently [24–27].

Similarly to sex-specific risk factors, factors that are associated with the persistence of ADHD have been poorly understood. Basically, research results have shown persistence rates of up to 40–60% across the life span [28–33] and symptomatic persistence rates of up to 86% [34]. This is in contrast to the earlier assumption that ADHD is an isolated childhood disease. While the hyperactive-impulsive symptoms tend to decrease over lifetime and then often present as inner restlessness, the inattentive symptoms tend to persist during adulthood more frequently [35,36]. In a large meta-analysis, severity of ADHD, treatment for ADHD, comorbid conduct disorder and comorbid major depressive disorder emerged as relevant predictors of ADHD persistence [37,38]. Additionally, a family history of ADHD [39] or having an affected first-degree relative [24] combine as relevant predictors. Remarkably, this list comprises predictors, not risk factors in the narrow sense. Knowledge about risk factors driving the persistence in ADHD has not markedly advanced since the analysis of the NCS-R data twenty years ago [10]. Among the exceptions is the finding from the English and Romanian Adoptees study according to which severe deprivation in early childhood is associated with the persistence of ADHD symptoms [14].

The lack of understanding of risk mechanisms regarding the persistence of ADHD is of special concern for women [40]: given that the male-to-female ratio in clinical ADHD samples changes from childhood (3:1) to adulthood (1:1) [41,42]. Hence ADHD persists more frequently in women than in men [43,44]. An exceptional finding reported by Biederman et al. [24] suggests that persistence in girls/ women is associated with family-related adversities such as lack of familial cohesion.

The present exploratory analyses addressed the research gaps by focusing on factors associated with different subgroups of ADHD: subthreshold (symptom level), full (diagnosis level) and in both subgroups persisting ADHD. The analyses were refined by implementing latent class analysis (LCA) in order to determine risk and associated factor related subgroups within subthreshold and full ADHD. The latent classes were expected to offer additional perspectives on the interplay between occurrence and persistence factors. Altogether the analysis design comprises four levels (including sex) and serves as a template for systematically approaching a heterogeneous disease target. Since preliminary analyses had suggested different patterns in women and in men, we first present here the results for women.

Analyses were based on data from a large Swiss population-based epidemiological cohort study, with information on a comprehensive array of factors emerging early in life and potentially being associated with ADHD. These include psychosocial factors such as family dysfunction, ACE and traumatic experiences. We also considered somatic diseases, which are mostly rooted in the first years of life such as childhood infections, allergic diseases and asthma, peptic ulcer and irritable bowel syndrome, and which imply an upregulation of immune system activity. In addition, we included immunological and cardiometabolic markers in the analyses, since we could recently show that shifts of inflammatory markers after traumatic experiences tend to persist

across the whole lifespan [45], which hypothetically applies to any developmental and mental disorder associated with ACE. Here again, the focus was on upregulated pro-inflammatory immune system activity.

## Materials and methods

### The CoLaus|PsyCoLaus cohort and instruments

Our data stemmed from CoLaus|PsyCoLaus, a randomly selected cohort from the population of the Swiss city of Lausanne (see also S1 Text/ Fig in S1 File). The CoLaus|PsyCoLaus cohort study was designed to decipher the relationship between mental disorders and cardiovascular diseases [46,47]. Recruitment went from 2003 to 2006. The physical baseline evaluation (CoLaus) went from May 2003 to June 2006, the psychiatric baseline evaluation (PsyCoLaus) went from January 2004 to May 2009. In Colaus, participants were followed up for the physical evaluations between April 2009 and September 2012, May 2014 and April 2017, April 2018 and March 2021, and have been followed since January 2022 for the fourth follow-up. In PsyCoLaus, participants were followed up for the psychiatric evaluations between April 2010 and January 2014, November 2014 and July 2018, and April 2018 and September 2021, and have been followed for the fourth time since October 2021.The data analysed here was based on the first survey and on follow-up 1 and follow-up 2 surveys; it consisted of 5111 participants, thereof 2741 women, with a baseline interview assessed within an age range of 35–88 years (S1 Text/ Fig in S1 File).

### Psychiatric and psychological assessments

The French version [48] of the semi-structured Diagnostic Interview for Genetic Studies (DIGS) [49] was used to assess diagnostic information on mental disorders. The DIGS collects information on a broad spectrum of the Diagnostic and Statistical Manual of Mental Disorders IV (DSM-IV) Axis I criteria related to mental disorders and, moreover, on the course and chronology of comorbid features [49]. Inter-rater and test-retest reliability of the French version were successfully established in a clinical sample in Lausanne for major mood and psychotic disorders [48] as well as for substance use disorders and antisocial personality [50].

The ADHD section of the questionnaire was adopted from the Yale Family Study version of the Schedule for Affective Disorders and Schizophrenia – Lifetime and Anxiety disorder version (SADS-LA) [51]. The ADHD section of this interview was developed in analogy to the corresponding section of the Kiddie-Schedule for Affective Disorders and Schizophrenia – Epidemiologic version (K-SADS-E) [52] and retrospectively assesses the occurrence of childhood ADHD. It was introduced by two probe questions anyone of which had to be answered positively to continue the interview in this section:

a) Was there ever a time in your childhood when you had trouble concentrating?

b) Was there ever a time in your childhood when you had problems because you acted before thinking or because you were too restless?

Separate subsections covered the inattentive section of ADHD in childhood (13 items), the hyperactive section (7 items) and the impulsive section (3 items). Each subsection recorded further information on ages of onset/ offset, and school/ familial/ social problems caused by the symptoms.

In the analysis, we only included subjects with an onset of symptoms before the age of 12 in accordance with the DSM-5 criteria for ADHD. Regarding subthreshold ADHD, we adapted the definition of Biederman et al. [53], however requiring the presence of at least three symptoms (instead of four) from either the inattentive or the hyperactive/ impulsive subgroup of symptoms. For an overview of the different definitions for subthreshold ADHD see Kirova et al. [54]. Persistence of ADHD was determined by continuingly experiencing symptoms beyond the age of 17 years, which is at the same time the cutoff age separating childhood and adulthood ADHD.

The familial aggregation of ADHD and other disorders was based on the semi-structured Family History–Research Diagnostic Criteria (FH-RDC) interview [55]. The latter comprised early and later anxiety disorders, mood and substance

use disorders (see S3 Text in S1 File) where their validity compared to the DIGS was extensively tested [56–58]. Further questionnaire and biomarker data used in the analyses covered the following issues: selected family related ACE based on questions from the SADS-LA [51], traumatic experiences reported in the post-traumatic stress disorder (PTSD) section of the DIGS [59], parental bonding using three subscales of the Parental Bonding Instrument (PBI) [60,61], somatic diseases assessed in the medical section of the DIGS or from physical assessments (CoLaus), blood/ urine derived physiological markers from CoLaus including white blood cells, cytokines, cardio-metabolic markers as well as cortisol variables [62]. The overview of the scales, variables and markers is provided in S2 Text in S1 File. Notably, certain instruments (FH-RDC, PBI) and physiological markers (e.g., cortisol measures) were only assessed in subsamples of PsyCoLaus for technical and practical reasons. This limited their use in clustering analyses (see below: latent class/ latent profile analysis).

## Ethics approval

The CoLaus|PsyCoLaus study was approved by the Institutional Ethics Committee for clinical research of the Medical and Biological Faculty of the University of Lausanne, which afterwards became the Ethics Committee of the Canton of Vaud for human research (www.cer-vd.ch; project number PB_2018–00038, reference 239/09). All participants received a detailed description of the goals, procedures and funding of the study and signed a written informed consent form. All procedures involved in this report comply with the ethical standards of the relevant national and institutional committees on human experimentation and with the Helsinki Declaration of 1975 (2008 revision) and in accordance with the applicable Swiss legislation.

## Data analyses

The analysis design included the following differentiations as shown in Fig 1:

- subthreshold vs. full ADHD;

- persistent vs. not persistent ADHD;

- subtypes/ subgroups of ADHD derived from latent class/ latent profile analysis (LCA/ LPA) which were based on selected risk and associated factors (i.e., familial, psychosocial and biological factors).

   The empirical aim of the study was to disentangle the association patterns between ADHD and a comprehensive array of factors/ covariates that change across these differentiation levels. In more technical terms, this study was based on a three-step strategy: a descriptive and exploratory step assessing the associations with psychosocial and biological factors in ADHD in women [63,64], a subgrouping step with LCA and an analysis step with distal variables.

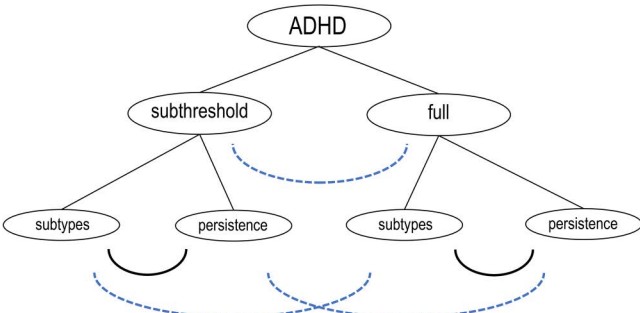

**Fig 1. Analysis design targeting subthreshold, full and persistent ADHD.**

Descriptive statistics were provided in terms of frequencies and means with standard errors. In analysis of associations with categorical variables, odds ratios were derived from contingency tables and χ2 test served to assess the p-values. In case of small cell frequencies the Fisher's exact test was applied. In analysis with metric variables (i.e., biomarker variables) we used the t-test. As we were focusing on association patterns across all analyses, applying adjustment for multiple testing was a contraindication (see also [65,66]).

Latent class/ latent profile analysis (LCA/ LPA) were conducted for subthreshold and full ADHD based on risk and associated factors with the aim to determine different ADHD subgroups. Like cluster analysis, LCA is a person-centered approach to classification, i.e., it aims to group individuals into homogeneous classes [67–69]. If it is based on metric instead of categorical variables, it is labelled latent profile analysis. In this study, we introduced variables of both types in the analysis, thus yielding a LCA/ LPA analysis and thus two different outcomes (probabilites for categorical variables and means for continuous variables).

The selection of variables entered into the LCA/ LPA was supposed to adhere to the unity of the matter which is crucial both for technical reasons (stability of the LCA) and theoretical reasons (interpretability of the latent classes). The focus is typically placed on variables that show a significant or trend-level association with ADHD, although this represents an imperfect criterion. Heterogeneity implies that associations of variables belonging to smaller classes may be suppressed, and these variables can therefore be overlooked during the selection procedure. Two basic strategies can be used to address this dilemma. The first borrows from a common variable-selection approach in regression analysis by setting a relatively lenient cutoff, for example $p < 0.25$, after initial screening [70]. The second strategy, applied here, is to begin the selection of variables based on a less lenient significance level, for example $p < 0.1$ and to balance this by subsequently reassessing the associations between the latent classes and variables initially not included in LCA/ LPA. In this study, categorical variables with cell frequencies $< 10$ (subthreshold ADHD) and $< 8$ (full ADHD) were omitted from the LCA as well as variables assessed only in subsamples (see above: FH-RDC, PBI).

Given that ADHD is a complex target, that LCA/ LPA applied on complex targets can behave in an erratic mode and that the ADHD subsamples are rather small, we used the LCA/ LPA in a pragmatic and rough way aiming:

a) to outline two to three latent classes that demarcate the most basic heterogeneity features of subthreshold and full ADHD; notably, in a theoretical perspective (and apart from technical restrictions and model fit measures in LCA) the enumeration of classes in LCAs of complex targets does not lead to any optimal number of classes per se; each solution, i.e., number of classes, delivers different valuable information;

b) to select and present a LCA/ LPA solution from the model space, which is plausible, well interpretable and towards which other examined models converge; in analogy to multiverse regression [71,72] and related applications, we assume also for each LCA a multiverse of models determined by acceptable variations of variable sets, variable definitions, aggregation levels etc.; however, so far, each reported LCA presents a bona fide solution;

c) to assess the associations between the latent classes and distal variables or variables excluded from the LCA/ LPA in a conventional "classify-analyse" way; given the lacking stability of LCAs in contexts shaped by a high level of complexity, we prefer a pragmatic approach instead of precision-like modeling in analyses with distal variables [73];

d) to assess associations with further ADHD specific variables such as the age of onset or persistence using the same strategy.

Further theoretical and methodological considerations regarding LCA/ LPA on complex targets are provided in S3 Text in S1 File.

The LCA/LPA was conducted within a similar framework as in other analyses of the group [66,74,75]. Akaike information criterion (AIC), the Bayesian information criterion (BIC), the sample-size adjusted BIC (ABIC), adjusted Lo-Mendell-Rubin likelihood ratio test (LMR-LRT) and the bootstraped likelihood ratio test (LRT) served as model fit

indices to evaluate the number of classes. Notably, the evaluation of modell fits from the "ICs" is different, i.e., less informative in an LPA and also in an LCA/ LPA than in a conventional LCA with exclusively categorical variables. Due to the limited ADHD sample size, up to four classes were evaluated regarding subthreshold ADHD level and up to three classes regarding full ADHD.

For data preprocessing and common analyses, SPSS version 28 was used. The LCA/ LPAs were carried out with Mplus Version 8.

## Results

Among the 2687 women who answered the two ADHD probe questions in at least one interview, 151 women (5.5%) were screened positively and continued to answer further questions of the ADHD section. The ADHD symptoms and their rates are listed in S4 Table in S1 File. Among them, 82 women (3.1%) met the subthreshold criteria, and 43 women (1.6%) met the criteria for full ADHD. In the subthreshold group, 36 women (41.9%) reported persisting symptoms; whereas in the ADHD group, 25 women (58.1%) reported persistance, respectively ($\chi^2 = 2.29$, d.f. 1, p = 0.13; OR=1.77, CI 0. 84–3.74).

The frequencies of associated factors by ADHD status are shown in Table 1 (categorical variables), and the descriptive parameters of the marker and further metric variables in Table 2 (z-standardized values) as well as in S5 Table in S1 File (raw values).

### Association patterns in subthreshold and full ADHD

Tables 1 and 2 reveal several similar association patterns in subthreshold and full ADHD:

- a consistently positive but moderate family history of mental disorders; an exception is family history of ADHD itself with an odds ratio higher than seven for full ADHD compared to a moderate value in subthreshold ADHD;

- positive and relatively high odds ratios for ACE ranging between 2.4 and 5.2;

- no associations with childhood infectious diseases and atopic diseases (except for drug allergy in subthreshold ADHD).

- a set of positive associations with comorbid somatic conditions, sometimes above and sometimes below the significance level, including gastro-intestinal conditions (peptic ulcer, irritable bowel syndrome);

- a consistent pattern of decreased parental bonding scores of care and encouragement of freedom and higher scores of denial of autnomy (only trend level for fathers in full ADHD);

In addition, high fever (typically referring to early and middle childhood, Table 1)) and lower blood pressure (BP) (Table 2) were specifically reported in subthreshold ADHD. In contrast to the latter, only trend level associations with cardiometabolic markers comprised increased triglyceride values in subthreshold ADHD (together with cortisol AUC increase) and increased body mass index (BMI) and HbA1c values in full ADHD.

Additional analyses directly comparing subthreshold and full ADHD reflected only the most marked differences, i.e., the high frequencies of ADHD familial aggregation as well as lower levels of parental bonding scores, in particular maternal care, in full compared to subthreshold ADHD (data not shown). Interestingly, the age of onset was reported to be distinctly lower in full than in subthreshold ADHD (4.53 vs. 6.31 years; (F = 24.82, d.f. 1, p = 0.000).

### Person-centered perspective on subthreshold and full ADHD based on LCA/ LPA

In subthreshold ADHD, according to the inclusion criteria, the LCA/ LPA analysis comprised the ACE variables, peptic ulcer, IBS, drug allergy, both BP parameters and triglycerides. The enumeration of the classes was stopped after a minor class emerged in the four class solution. The model with three classes was adopted given balanced class sizes, a convenient configuration of fit indices and well interpretable classes. The fit indices of the consecutive models are tabulated in

**Table 1. Risk/ associated factors in subthreshold and full ADHD in women (part 1: categorical variables). Frequencies and odds ratios (OR) with confidence intervals (CI).**

| | subthreshold ADHD | | OR (CI) | p-value | full ADHD | | OR (CI) | p-value |
|---|---|---|---|---|---|---|---|---|
| | yes (n = 82) | no (n = 2536) | | | yes (n = 43) | no (n = 2541) | | |
| | n (%) | n (%) | | | n (%) | n (%) | | |
| familial aggregation | | | | | | | | |
| ADHD | 5 (8.3) | 72 (4.5) | 1.92 (0.75-4.95) | .167 | 10 (26.3) | 72 (4.5) | **7.56 (3.54-16.17)** | **.000** |
| early anxiety disorders[1] | 37 (59.7) | 599 (35.5) | **2.69 (1.60-4.51)** | **.000** | 19 (50.0) | 599 (35.5) | *1.81 (0.95-3.45)* | *.066* |
| later anxiety disorders[2] | 15 (24.2) | 226 (13.4) | **2.06 (1.13-3.74)** | **.016** | 7 (18.4) | 226 (13.4) | 1.46 (0.63-3.35) | .372 |
| mood disorders[3] | 45 (66.1) | 761 (45.2) | **2.37 (1.39-4.04)** | **.001** | 24 (63.2) | 762 (45.2) | **2.08 (1.07-4.04)** | **.028** |
| substance use disorders[4] | 9 (14.5) | 192 (11.4) | 1.32 (0.64-2.72) | .450 | 8 (21.1) | 192 (11.4) | *2.07 (0.94-4.59)* | *.066* |
| childhood adversities and trauma | | | | | | | | |
| inter-parental violence | 25 (30.5) | 301 (12.0) | **3.19 (1.97-5.19)** | **.000** | 17 (41.5) | 301 (12.0) | **5.18 (2.75-9.75)** | **.000** |
| fear of parental maltreatment | 17 (20.7) | 244 (9.7) | **2.43 (1.40-4.22)** | **.001** | 13 (30.2) | 244 (9.7) | **4.03 (2.07-7.83)** | **.000** |
| trauma below age of 10 | 13 (16.0) | 110 (4.3) | **4.21 (2.26-7.84)** | **.000** | 6 (14.0) | 111 (4.4) | **3.54 (1.47-8.57)** | **.003** |
| childhood infectious diseases | | | | | | | | |
| pertussis | 4 (4.9) | 255 (10.1) | 0.46 (0.17-1.26) | .122 | 3 (7.0) | 255 (10.0) | 0.67 (0.21-2.19) | .507 |
| rubella | 7 (8.5) | 204 (8.0) | 1.01 (0.49-2.35) | .872 | 4 (9.3) | 205 (8.1) | 1.17 (0.41-3.30) | .768 |
| chickenpox | 66 (84.6) | 1995 (85.3) | 0.95 (0.51-1.77) | .861 | 34 (89.5) | 1997 (85.3) | 1.46 (0.52-4.14) | .474 |
| measles | 57 (75.0) | 1811 (78.5) | 0.82 (0.48-1.39) | .461 | 33 (84.6) | 1813 (78.6) | 1.50 (0.63-3.60) | .360 |
| mumps | 50 (64.9) | 1363 (58.6) | 1.31 (0.81-2.11) | .266 | 26 (63.4) | 1364 (58.6) | 1.22 (0.64-2.32) | .536 |
| herpes simplex | 15 (18.3) | 483 (19.2) | 0.95 (0.54-1.67) | .844 | 12 (27.9) | 483 (19.2) | 1.63 (0.83-3.21) | .149 |
| allergic diseases and atopies | | | | | | | | |
| asthma | 12 (14.6) | 330 (13.0) | 1.15 (0.61-2.14) | .668 | 4 (9.3) | 331 (13.0) | 0.69 (0.24-1.93) | .471 |
| hay fever | 19 (23.2) | 453 (17.9) | 1.39 (0.82-2.34) | .218 | 8 (18.6) | 453 (17.8) | 1.05 (0.48-2.29) | .895 |
| eczema | 8 (9.8) | 297 (11.7) | 0.82 (0.39-1.71) | .587 | n<3 | | | |
| drug allergy | 16 (19.5) | 308 (12.1) | **1.75 (1.00-3.07)** | **.046** | 6 (14.0) | 308 (12.1) | 1.18 (0.49-2.81) | .715 |
| comorbid diseases/ conditions | | | | | | | | |
| high fever | 9 (11.3) | 139 (5.7) | **2.08 (1.02-4.24)** | **.040** | 3 (7.1) | 140 (5.8) | 1.25 (0.38-4.10) | .709 |
| peptic ulcer | 10 (12.2) | 110 (4.3) | **3.06 (1.54-6.10)** | **.001** | 8 (18.6) | 110 (4.3) | **5.05 (2.29-11.15)** | **.000** |
| irritable bowel syndrome | 10 (12.2) | 122 (4.8) | **2.75 (1.38-5.46)** | **.003** | n<3 | | | |

Key: *italic: p < 0.1*, **bold: p < 0.05**, ***italic & bold: p < 0.01.***

Notes:

1 Separation anxiety disorder, overanxious disorder, specific phobias (animals), social phobia.

2 Generalized anxiety disorder, panic, agoraphobia, specific phobias (excl. animals).

3 Major depression disorder, dysthymia, bipolar disorders.

4. Alcohol, cannabis, other illicit drugs abuse/dependence.

*General remark: percentages can slightly deviate due to different numbers of missing values in different variables.*

S6 Table in S1 File. The probabilities (categorical variables) and means (metric variables) of the three class solution are shown in Fig 2. Both classes 1 and 3 were characterized by increased probabilities of ACE, however in class 1 this was combined with decreased BP levels (N = 30, 37.0%; labelled "ACE low BP") and in class 3 with increased BP and triglyceride levels (N = 11, 13.6%; labelled "ACE high BP"). While IBS was more prominent in class 1 than in classes 2 and 3, other variables (peptic ulcer, drug allergies) showed no conspicuous variation across the classes. Class 2, the biggest class (N = 40, 49.4%), had no noteworthy features and therefore represented an "idiopathic class" where no risk/ associated factors were prominent.

**Table 2. Risk/ associated factors in subthreshold and full ADHD in women (part 2: metric variables). Z-standardized variables with mean for ADHD, standard errors (SE).**

| | | subthreshold ADHD | | | full ADHD | |
|---|---|---|---|---|---|---|
| | n/ total n | mean (SE) | p-value | n/ total n | mean (SE) | p-value |
| leukocytes | 75/ 2349 | −0.010 (0.094) | .953 | 33/ 2408 | 0.079 (0.186) | .602 |
| basophils | 74/ 2348 | −0.072 (0.101) | .462 | 32/ 2406 | 0.052 (0.167) | .745 |
| eosinophils | 74/ 2338 | 0.047 (0.113) | .661 | 32/ 2396 | 0.037 (0.190) | .823 |
| lymphocytes | 75/ 2346 | 0.013 (0.091) | .862 | 32/ 2404 | 0.177 (0.163) | .227 |
| monocytes | 75/ 2342 | −0.136 (0.094) | .179 | 32/ 2397 | −0.076 (0.152) | .611 |
| neutrophils | 75/ 2340 | −0.024 (0.088) | .831 | 32/ 2398 | 0.011 (0.223) | .934 |
| interleukin 6 | 80/ 2577 | 0.030 (0.084) | .764 | 43/ 2650 | 0.098 (0.119) | .439 |
| interleukin 1β | 80/ 2579 | 0.002 (0.091) | .991 | 43/ 2652 | −0.055 (0.120) | .631 |
| TNF-α | 80/ 2574 | 0.029 (0.067) | .723 | 43/ 2647 | −0.032 (0.114) | .755 |
| hsCRP | 81/ 2592 | −0.078 (0.084) | .434 | 43/ 2665 | 0.154 (0.131) | .265 |
| BMI | 82/ 2607 | −0.029 (0.114) | .829 | 43/ 2681 | 0.273 (0.126) | *.052* |
| waist-hip ratio | 82/ 2615 | 0.007 (0.094) | .891 | 43/ 2689 | 0.133 (0.126) | .260 |
| systolic blood pressure | 82/ 2617 | −0.163 (0.096) | **.048** | 43/ 2691 | 0.093 (0.129) | .461 |
| diastolic blood pressure | 82/ 2617 | −0.216 (0.113) | **.017** | 43/ 2690 | 0.112 (0.143) | .404 |
| cholesterol total | 82/ 2614 | 0.126 (0.080) | .140 | 43/ 2687 | 0.144 (0.132) | .223 |
| HDL cholesterol | 82/ 2616 | −0.032 (0.011) | .727 | 43/ 2690 | −0.059 (0.150) | .652 |
| LDL cholesterol | 82/ 2613 | 0.124 (0.084) | .159 | 43/ 2687 | 0.140 (0.150) | .250 |
| triglycerides | 82/ 2614 | 0.143 (0.095) | *.097* | 43/ 2688 | 0.195 (0.108) | .107 |
| insulin | 81/ 2568 | −0.027 (0.093) | .781 | 43/ 2641 | −0.026 (0.113) | .843 |
| glucose | 82/ 2610 | −0.004 (0.072) | .949 | 43/ 2675 | 0.082 (0.105) | .442 |
| HbA1c | 67/ 2102 | 0.001 (0.103) | .995 | 30/ 2155 | 0.256 (0.159) | *.055* |
| adiponectin | 80/ 2550 | 0.069 (0.100) | .455 | 43/ 2623 | 0.095 (0.123) | .448 |
| leptin | 77/ 2528 | 0.039 (0.113) | .687 | 43/ 2601 | −0.040 (0.141) | .788 |
| salivary cortisol | | | | | | |
| awakening response | 47/ 1453 | −0.113 (0.124) | .735 | 24/ 1625 | 0.088 (0.206) | .673 |
| AUC/ ground | 52/ 1651 | −0.091 (0.131) | .424 | 24/ 1625 | 0.144 (0.197) | .400 |
| AUC/ increase | 53/ 1647 | 0.177 (0.090) | *.099* | 24/ 1620 | 0.051 (0.195) | .746 |
| diurnal cortisol slope | 51/ 1649 | −0.133 (0.101) | .271 | 23/ 1622 | 0.015 (0.137) | .945 |
| PBI mother | | | | | | |
| care | 49/ 1480 | −0.275 (0.148) | **.035** | 30/ 1527 | −0.763 (0.181) | ***.000*** |
| denial of autonomy | 49/ 1480 | 0.361 (0.149) | ***.007*** | 30/ 1527 | 0.613 (0.177) | ***.000*** |
| encouragement of freedom | 49/ 1480 | −0.322 (0.163) | **.015** | 30/ 1527 | −0.716 (0.188) | ***.000*** |
| PBI father | | | | | | |
| care | 46/ 1421 | −0.263 (0.157) | *.059* | 26/ 1464 | −0.580 (0.212) | ***.003*** |
| denial of autonomy | 46/ 1421 | 0.422 (0.166) | ***.003*** | 26/ 1464 | 0.363 (0.250) | *.062* |
| encouragement of freedom | 46/ 1421 | −0.407 (0.156) | ***.004*** | 26/ 1464 | −0.573 (0.233) | ***.003*** |

Key: *italic: p<0.1*, **bold: p<0.05**, ***italic & bold: p<0.01.***

Notes:

•reference category typically with mean>−0.005 and <0.005; SE<0.02.

•total n in analyses with subthreshold ADHD does not comprise participants with full ADHD (and vice versa).

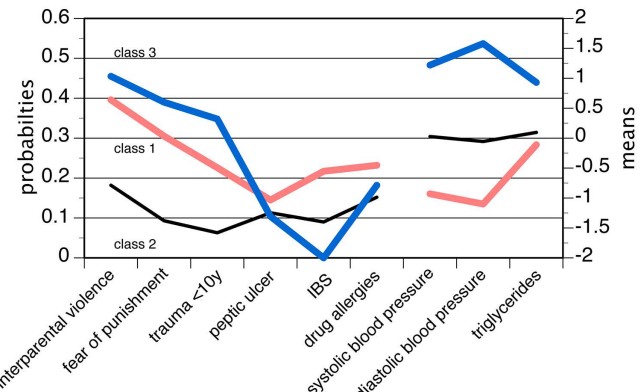

**Fig 2. Latent class/ latent profile analysis of factors associated with subthreshold ADHD in women.** Means (first panel) and probabilities (second panel) of the three-class model; the means and probabilities of the items were connected by lines in order to facilitate examination of the LCA/ LPA.

In a reanalysis of the association patterns, BMI (F = 4.37, d.f. 2, p = 0.016), glucose (F = 4.66, d.f. 2, p = 0.012) and insulin (F = 4,82 d.f. 2, p = 0.011) matched with the three class LCA/ LPA, in particular by sharing high levels with the "ACE high BP " class. Among the familial aggregation variables, no disorders stood out. The PBI subscales did not show any conspicuous variation between the classes. The age of onset was slightly lower in class 3 (5.18 years) than in class 1 (6.60 years) or class 2 (6.26 years), but the differences were not statistically significant ((F = 1.61, d.f. 2, p = 0.206).

According to the inclusion criteria, the LCA/ LPA of full ADHD comprised the ACE variables, peptic ulcer and BMI. We adopted the solution with two classes, again considering the class sizes, the configuration of fit indices and the interpretability of the classes. The fit indices of the consecutive models are tabulated in S7 Table in S1 File. Fig 3 shows the probabilities (categorical variables) and means (metric variables) of the two-class model. The larger group (Latent Class 1: N = 26, 63.4%) in the two-class model represented an "idiopathic class" without any noteworthy distinctive features. In contrast, the smaller group (Latent Class 2: N = 15, 36.6%), labelled as the "burdened class", was characterized by increased probabilities for ACE together with peptic ulcer and increased levels of BMI.

The reanalysis of the association patterns pointed at higher insulin levels in the "burdened class" and lower levels in the "idiopathic class" (F = 5.95, d.f. 1, p = 0.020). Among the PBI variables, the mother's care subscale showed markedly

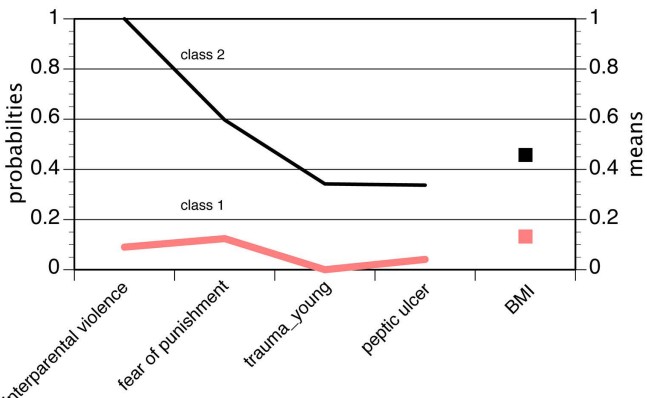

**Fig 3. Latent class/ latent profile analysis of factors associated with full ADHD in women.** Means (first panel) and probabilities (second panel) of the two-class model; the means and probabilities of the items were connected by lines in order to facilitate examination of the LCA/ LPA.

lower values in the "burdened class" (F = 9.21, d.f. 1, p = 0.005). Among the familial aggregation variables, both early and late anxieties were involved with higher proportions in the "burdened class" ($\chi^2$ = 5.39, d.f. 1, p = 0.020 and $\chi^2$ = 5.98, d.f. 1, p = 0.014). The familial aggregation of ADHD did not differ between the classes. No further variables contributed to a better differentiation between the two classes, and also the ages of onset were similar (4.58 vs. 4.47 years; F = 0.28, d.f. 1, p = 0.796).

### Analyses related to persistence of subthreshold and full ADHD

In subthreshold ADHD, persistence was positively associated with high fever and herpes simplex, and also with BP variables (Table 3, variable-centered approach). The classes derived from LCA/ LPA showed a gradient for persistence ($\chi^2$ = 8.76, d.f. 2, p = 0.013; see Table 4). The lowest proportion of persistent ADHD emerged in the "ACE low BP" class (class 1: 20.8%), and the highest proportion in the "ACE high BP" class (class 3: 76.9%). The "idiopathic" class was positioned in between (47.7%). Ages of onset according to persistence (5.8 years vs. 6.6 years) slightly differed with a trend significance p-value (F = 2.54, d.f. 1, p = 0.11).

For full ADHD, persistent ADHD was associated with drug allergy and peptic ulcer (Table 5). Furthermore, increased levels of immunological markers (increased monocyte, neutrophil, and hsCRP levels) became the focus of attention in addition to cardiometabolic markers (insulin, leptin).

Examining persistence across LCA/ LPA derived classes (Table 6) showed that the proportion of persistent full ADHD was potentially higher in the "burdened" class than in the "idiopathic" class, but did not reach the level of statistical

**Table 3. Summary of analyses of risk/ associated factors and persistence in subthreshold ADHD in women beyond age of 17 (specific/ significant associations).**

| specific associations | persistence (n = 36) | no persistence (n = 46) | p-value Fisher's exact test |
|---|---|---|---|
| high fever | 9 (25.0%) | 0 (0%) | *.000* |
| herpes simplex | 11 (30.6%) | 4 (8.7%) | **.019** |
| | | | p-value |
| | mean (SE) | mean (SE) | t-test |
| systolic blood pressure | 0.091 (0.128) | −0.361 (0.133) | **.019** |
| diastolic blood pressure | 0.169 (0.157) | −0.517 (0.146) | *.002* |

Note: marker variables are z-standardised.

Key: **bold: p < 0.05**, *italic & bold: p < 0.01.*

**Table 4. Summary of analyses of risk/ associated factors and persistence in subthreshold ADHD in women beyond age of 17 (latent classes).**

| associations with latent classes | persistence (n = 36) | no persistence (n = 45) |
|---|---|---|
| class 1 (row %) | 9 (30.0%) | 21 (70.0%) |
| class 2 (row %) | 18 (45.0%) | 22 (55.0%) |
| class 3 (row %) | 9 (81.8%) | 2 (18.2%) |

Note: chi-square = 8.76, d.f. 2, p-value = .013.

class1: high ACE burden, low blood pressure.

class2: "idiopathic".

class3: high ACE burden, high blood pressure and increased levels in other cardio-metabolic markers.

**Table 5. Summary of analyses of risk/ associated factors and persistence in full ADHD in women beyond age of 17 (specific/ significant associations).**

| specific associations | persistence (N = 25) | no persistence (N = 18) | p-value Fisher's exact test |
|---|---|---|---|
| drug allergy | 6 (24.0%) | 0 (0%) | **.032** |
| peptic ulcer | 8 (32.0%) | 0 (0%) | **.013** |
| | | | p-value |
| | mean (SE) | OR (CI) | t-test |
| monocytes | 0.259 (0.165) | **3.7 (1.2-11.7)** | **.010** |
| neutrophils | 0.394 (0.328) | **2.0 (1.0-4.0)** | **.050** |
| hsCRP | 0.396 (0.158) | **2.4 (1.1-5.4)** | **.028** |
| insulin | 0.179 (0.153) | **2.9 (1.0-8.4)** | **.035** |
| leptin | 0.283 (0.193) | ***3.0 (1.3-7.2)*** | ***.005*** |

Note: marker variables are z-standardised.

Key: **bold: p < 0.05**, ***italic & bold: p < 0.01.***

**Table 6. Summary of analyses of risk/ associated factors and persistence in full ADHD in women beyond age of 17 (latent classes).**

| associations with latent classes | persistence (N = 25) | no persistence (N = 18) |
|---|---|---|
| class 1 (row %) | 12 (46.2%) | 14 (53.8%) |
| class 2 (row %) | 11 (73.3%) | 4 (26.7%) |

Note: chi-square = 2.85, d.f. 1, p-value (chi-square test) =.091.

class1: "idiopathic".

class2: high ACE burden.

significance. Moreover, age of onset was slightly lower in persistent full ADHD (4.1 years vs. 4.6 years) but also did not reach the 0.05-significance level (F = 1.41, d.f. 1, p = 0.24).

Additional analyses on persistence involving both subthreshold and full ADHD – and thus based on higher figures – focused on the sum score of ADHD symptoms and on familial aggregation of ADHD. Women who reported persisting ADHD had a slightly higher mean ADHD symptom score ((8.0 vs. 8.9), but the difference was not significant at the 0.05-level (F = 1.51, d.f. 1, p = 0.22). Persistence of ADHD was also slightly more frequent in women who reported familial aggregation of ADHD 66.6% vs. 45.8% (($\chi^2$ = 2.22, d.f. 1, p = 0.14; OR= 2.39, CI 0.75–7.53).

In sum, the probability of persistence of ADHD in women is typically associated with a combination of burden (such as ACE) and additional factors, here emerging from dysregulation within the cardio-metabolic and immunological systems. The associations with ADHD severity and familial aggregation are less resounding although they show in the expected direction.

## Discussion

This study examined risk factors, early emerging somatic comorbidities and physiological markers that are potentially associated with the occurrence and persistence of ADHD in women. It focused on association patterns across different subgroups, i.e., subthreshold vs. full ADHD, persisting vs. non-persisting ADHD and subgroups determined by LCA/ LPA. This comparative analysis design aimed to grasp the heterogeneity of ADHD as a complex disorder and presents a

template of how to assess the full picture of association patterns across different disorder levels (subthreshold, full, persistent) by applying both variable- and person-centered approaches.

We observed dominating subgroups determined by LCA/ LPA in both ADHD subforms: the one related to ACE and the other lacking any specific risk factors or correlates. The former subgroup was labelled as "burdened" (in full ADHD) or "ACE high BP"/ "ACE low BP" (in subthreshold ADHD), whereas the latter was labeled as the "idiopathic" subgroup. Moreover, in both ADHD subforms (full and subthreshold), persistence was associated with the "burdened" subgroups, notably if accompanied by additional features, i.e., persisting dysregulation of immunological and cardio-metabolic parameters as also seen in PTSD [45]. Complementary to these dominating subgroups, full ADHD differed from subthreshold ADHD in terms of a strongly increased impact of familial aggregation and a lower age of onset. The proportion of persistent ADHD between full and subthreshold was slightly increased, but the difference was in a smaller range than between the LCA/ LPA subgroups within each subform (full and subthreshold).

### Risk factors and associated factors in subthreshold and full ADHD in women

Among the wide range of factors associated with ADHD shown in previous research [76] and included in the present analyses, ACE and parental bonding stood out. The basic impact of ACE on ADHD has been thoroughly demonstrated in previous research based on many and diverse heterogeneous items including abuse and maltreatment, neglect, foster placement, domestic violence, familial substance use, familial mental illness, familial death, familial criminality, parental separation/ divorce, bullying, socioeconomic adversity, neighbourhood violence and others [17,33,77–80]. This is despite the fact that ACE undoubtedly play a double causal role on ADHD pathogenesis and reciprocally on ADHD trajectories [12,81]. The strength of the associations with ADHD varies between the listed factors, and it varies in parallel with their number, i.e., along a dose-response relationship [80,82]. Among the most conspicuous studies are the Romanian orphan studies, which not only documented the disastrous impact of early life deprivation but also provided temporal information about the critical age period – up to the age of two – when more severe forms of ADHD are formed [15,83,84]. However, ACE are predictive of a bulk of heterogeneous neurodevelopmental and common mental disorders apart from ADHD [85]. This multifinality suggests a comprehensive impact on brain development and functioning [86]. An in-depth understanding of the mutifinality of ACE is still desperately needed.

In this study, the ACE items were selected to represent severe forms of adverse experiences, which can be expected to yield the strongest associations with ADHD [80]. These are manifest forms of violence both between the parents and against the child, which are close to traumatic experiences and, in case of interparental violence, leave no open questions about the direction of causal effects. All these items yielded stronger associations in full than in subthreshold ADHD.

Another mutual marker across different levels of ADHD in this study was peptic ulcer. This came as a surprise, since peptic ulcer has not been mentioned regarding ADHD so far. However, associations with peptic ulcer disease have already been found in several psychiatric conditions such as depression and generalized anxiety disorder [87]. Some older studies also suggested an association between peptic ulcers and psychiatric conditions in childhood [88]. Again there is potential for controversy such as regarding ACE, since peptic ulcer disease might hypothetically be considered as a consequence of stressful experiences or risky behavior due to ADHD instead of as an early surrogate of stress, although there is much evidence that psychological stress increases the risk for peptic ulcers [89,90]. In addition, we found that peptic ulcer was not a universal feature of ADHD but was involved in the "burdened" subgroup in LCA/LPA (see below) and thus most probably represented a somatic indicator of early stress exposure.

In contrast to mutual markers across different levels of ADHD, familial aggregation [12] was of particular importance for full ADHD. In subthreshold ADHD, low BP [91,92] was a unique feature. Altogether, this suggests that the differentiation between subthreshold and full ADHD in women does not only depend on dose-response relationships among risk factors, for example the severity of ACE, but also involves additional features.

Consistent with former research [54] we found earlier ages of onset in full ADHD than in subthreshold ADHD, whereas age of onset played a less important role with regard to ADHD persistence. It has been suggested that age of onset in ADHD mirrors an accumulation of environmental and genetic risk factors [13,54]. Interestingly, we found a similar phenomenon in stuttering: earlier age of onset in persistent than in non-persistent stuttering [93]. However, comparisons with age of onset in stuttering and in other neurodevelopmental disorders (e.g., autism spectrum disorder, Tourette syndrome) suggest that the sequence of ages of onset might reflect how risk mechanisms interfere at different stages of brain development. This interpretation is further corroborated by the remarkable finding that different factors are associated with persistence in full and in subthreshold ADHD.

## Subgrouping along risk and other associated factors in ADHD in women

Several subgrouping/ subtyping efforts have already tackled heterogeneity in ADHD beyond the conventional subtyping of ADHD into "inattentive", "hyperactive/impulsive", and "combined". They have covered different research domains: comorbidity [94], temperament [95], neuropsychological measures [96], EEG and ECG [97]. In addition, several studies have derived subgroups based on LCA of childhood risk factors [18,98,99] using a whole range of ACE representing different dimensions and risk mechanisms (as discussed above). This typically revealed three or four latent classes reflecting a low-high ACE burden pattern with intermediate classes depending on the selection of ACE variables and their mixtures of biologocial, psychological and sociological backgrounds. This is illustrated by an outstanding Swedish register study with six classes based on over 10,000 adolescents [100].

Another outstanding study is from Nguyen et al. [18], since it focused on women with and without ADHD. It used ten ACE items to derive four common ACE classes as found in other studies [75], labeled by the authors as "low exposure", "familial dysfunction", "emotional maltreatment" and "pervasive exposure". The authors only found only a conspicuous association between ADHD and the "emotional maltreatment" class.

In our study, we specifically examined latent classes and different factors in women with ADHD. Based on preliminary association analyses related to subthreshold and full ADHD in women, we derived basic ADHD classes – labeled as "idopathic" and "burdened". The burdened class comprised the majority of risk factors in addition to familial aggregation. In the burdened classes, the ACE variables such as interparental violence, fear of parental maltreatment, experience of trauma in childhood predominated together with cardio-metabolic markers. This is also in line with previous research on ACE in ADHD [10,17,79]. Moreover, the results indicate that severe psychosocial stress in childhood is relevant both with regard to full and to subthreshold ADHD.

Along with ACE, cardio-metabolic markers emerge as conspicuous factors both in full and in subthreshold ADHD. While interfering levels of BMI or triglycerides might be expectable in view of former research [40], the interpretation of the ambiguous role of BP variables in subthreshold ADHD (low levels despite high ACE in class 1 vs. high levels with high ACE in class 3) is challenging.

Two observations deserve particular attention beyond this initial comment. The "idiopathic" subgroup remained as a marked blind spot both in subthreshold and in full ADHD. This phenomenon occurs similarly in most other neurodevelopmental and psychiatric diseases. Second, familial aggregation was associated with both major subgroups in ADHD in women in reassessment analyses after conducting LCA/ LPA. Thus, familial aggregation – like other variables with minor variation between the classes – emerged as an overall associated factor.

## Risk and associated factors for ADHD persistence in women

Factors related to persistence of ADHD in women turned out to comprise conspicuous differences compared to occurrence. Severity of ADHD [37,101], familial aggregation [39] or the parental bonding dimensions from the PBI contributed in a moderate manner at best. No specific ACE stood out, however, persistence was more frequent in the burdened than

in the idiopathic subgroups both in full and in subthreshold ADHD, thus suggesting that several ACE or their combinations instead of any specific ACE would have a greater impact on persistence [82].

Moreover, dysregulation within cardio-metabolic and immunological systems came to the fore in associations with persistence of ADHD in women. Among cardiometabolic markers, BP stood out in subthreshold ADHD whereas increased insulin and leptin levels were conspicuous in full ADHD. The role of lower BP in subthreshold ADHD is challenging once more, given its inverse association with persistence (lower BP = less persistence) and its hypothesized link to hypoarousal and a dysregulated autonomic nervous system [92,102]. Notably, the association was not confounded by stimulant medication, which was not available to participants in the CoLaus|PsyCoLaus during childhood and adolescence. Together, these observations raise the possibility that lower BP may reflect a distinct subtype of subthreshold ADHD, potentially characterized by altered arousal or autonomic regulation.

As shown by endocrinological research [103–106], leptin has systemic effects on the regulation of neuroendocrine and immune functions. Leptin expression in humans is correlated with insulin levels. Insulin and leptin resistance have already been explored in other mental disorders [107] but rarely in studies on ADHD [108]. In full ADHD, also monocytes, neutrophils and hsCRP were associated with the persistence of ADHD in women. Based on recent research showing that leukocyte levels remain continuingly upregulated after traumatic experiences in childhood in PTSD [45], we hypothesize that the same phenomenon is apparent in ADHD. The proinflammatory state plays a crucial role in the pathogenesis of ADHD as for other psychiatric disorders [109,110] rather than being a consequence of ADHD. Notably, the positive association of persistence of ADHD with monocyte levels might potentially be related to the stress-mediated, catecholamine-induced activation of the relevant immune cells [109].

Summarizing, this study showed that not all girls with ADHD have the same probability that ADHD persists. Persistence has emerged as a topic that is no less important for understanding ADHD outcomes than the occurrence of ADHD itself. Beyond modest effects of known factors such as severity and familial aggregation [38,111], persistence appears to be linked to a specific subgroup which is burdened by psycho-social factors such as ACE and traumatic experiences and, in addition, if cardio-metabolic and inflammatory processes are upregulated and a proinflammatory state is retained. These factors and processes introduce a challenging new perspective, as all of them represent potential targets for intervention. Moreover, disentangling the heterogeneity of ADHD may provide important clues for applying interventions in a subgroup-specific manner. Research in this direction is ongoing [112] and may further benefit from parallel efforts in studies of other disorders [113–115].

## Methodological remarks

This study featured two approaches to deal with ADHD as a complex target. First, it aimed to achieve comparative results across different ADHD levels (subthreshold, full, persistence in both subforms) and across different analysis strategies (variable- and person-centered analyses).

Second, this study narrowed its analytical focus on patterns and pattern recognition. Within a multisystem framework of medium to high complexity, as in ADHD, conventional scientific principles such as precision and parsimony cease to operate appropriately at the level of elementary findings. Instead, patterns serve as the preliminary focus for disentangling which systems contribute to the complex target and how they interact. Every analysis in this study, whether explicitly (LCA) or implicitly, contributed to the identification, elaboration, reproducibility, and interpretability of patterns.

## Limitations

The results of this study should be interpreted in the light of several limitations. First, this is an exploratory study aiming to disentangle the heterogeneity of ADHD, in order to enable appropriate differentiation between relevant processes and epiphenomena in future studies. Important perspectives on ADHD heterogeneity are not addressed in the present report

– including ADHD in men and comorbidities with other neurodevelopmental and mental disorders – to avoid an overload of information. These aspects will be addressed in subsequent work.

Second, the initial association analyses included a broad range of protentially relevant variables from the CoLaus|Psy-CoLaus cohort, which have been also used in former studies of our group [93,116]. However, there are ADHD relevant variables that are not available in CoLaus|PsyCoLaus – for example prenatal and obstetric problems, exposure to smoking in parents – and could not be included in the present analyses.

Third, focusing on association patterns virtually excludes conclusions about causal mechanisms, i.e., risk factors in the narrow sense, even in cases when the evidence from a comparative and theoretical perspective is overwhelming. The impact of childhood adversities is the most prominent issue, for example the impact of inter-parental violence on risk for ADHD, stuttering, conduct disorder and many other early and late mental disorders. Remarkably, the multifinality of this impact excludes reverse causality, and similarly a mutual underlying factor. Given that a general mechanism on how to include comparative and theoretical information in discussing associations and their patterns is lacking, the conclusions also in this study necessarily remain incomplete.

Fourth, information related to symptoms, comorbid diseases/ disorders and drugs used for therapy was obtained retrospectively and may be subject of recall bias. The participants in PsyCoLaus were aged 35–88 years at the time of first interview which reinforces recall bias regarding symptoms and other issues during childhood and adolescence. Thus, this study is generally shifted towards more burdening symptoms and perceptions, which gives the impression of an extreme-group design that implies an elevated cut-off between subthreshold and full ADHD. In addition, underreporting of symptoms might include a gender-specific bias: girls/ women might be more prone than boys/ men to try to hide their symptoms of ADHD in order to avoid conflicts or might be more vulnerable to the development of self-blaming attributions [117,118].

Fifth, underreporting of persistence might occur since many affected persons no longer fulfil the full diagnostic criteria in adulthood and/ or might mask their symptoms through learned compensatory mechanisms. Several studies have also indicated that adults with ADHD tend to underestimate their own ADHD related impairments and show a low level of ADHD awareness when using a self-report screening questionnaire [119,120].

Finally, there are some technical limitations. Subgrouping of women with persistent ADHD could not be accomplished in this study due to sample size restrictions. While the results suggest that different mechanisms may contribute to the persistence of full ADHD in women – for example, pro-inflammatory processes beyond symptom severity – no conclusions can yet be drawn as to whether these mechanisms apply to all women or to specific subgroups.

Furthermore, inaccurate recall might affect reported age of onset in connection with the telescoping effect [121]. Telescoping implies that a remote onset age is typically advanced on the time scale by up to 3 years.

## Conclusions

This study highlighted a full picture of subthreshold, full and persistent ADHD in women with a broad range of risk and associated factors. The association patterns turned out to be heterogeneous across all levels. The most constant factors were ACE, notably with different association strengths (subthreshold vs. full ADHD) and along different accumulation patterns (individual variables vs. latent classes in persistent ADHD, both subforms). Beyond that, factors which are associated with occurrence and with persistence of ADHD, remarkably differ. In persisting full ADHD, altered immunological and cardio-metabolic profiles were identified, as indicated by lastingly increased levels of monocytes, neutrophils, hsCRP, insulin and leptin. This configuration ressembles the one in PTSD due to traumatic experiences in childhood. In persisting subthreshold ADHD, BP added to the heterogeneity: high BP as a risk factor and low BP as a protective factor. Persisting ADHD in women seems to be related to additional physiological systems that are involved during the development of the disorder and in its aftermath.

## Supporting information

**S1 File. S1 Text/ Fig: The CoLaus|PsyCoLaus cohort.** S2 Text: Psychiatric, psychological and somatic assessments. S3 Text: Theoretical and methodological considerations in LCA/ LPA on complex targets. S4 Table: Retrospectively reported childhood ADHD symptoms in women. S5 Table: Raw values of marker variables by measurement, overall sample, women. S6 Table: Subthreshold ADHD in women: model fit indices in LCA/ LPA, classes 1–4. S7 Table: Full ADHD in women: model fit indices in LCA/ LPA, classes 1–3. S8 Text: References. S9 Table: Low-level aggregate data (examples). (ZIP)

## Acknowledgments

The authors would like to express their gratitude to the Lausanne inhabitants who volunteered to participate in the CoLaus|PsyCoLaus cohort. The authors are grateful to Dr Stephanie Rodgers for her seminal contributions which paved the way for this study.

## Author contributions

**Conceptualization:** Ana Buadze, Vladeta Ajdacic-Gross, En-Young N. Wagner, Roland von Känel, Jennifer Glaus, Caroline L. Vandeleur.

**Data curation:** Enrique Castelao, Marie-Pierre F. Strippoli.

**Formal analysis:** Vladeta Ajdacic-Gross, Mario Müller, Setareh Ranjbar, Enrique Castelao, Marie-Pierre F. Strippoli.

**Funding acquisition:** Martin Preisig.

**Investigation:** Ana Buadze, Yanhua Xu, En-Young N. Wagner.

**Methodology:** Vladeta Ajdacic-Gross, Mario Müller, Yanhua Xu, Setareh Ranjbar.

**Project administration:** Martin Preisig, Caroline L. Vandeleur.

**Resources:** Erich Seifritz, Martin Preisig.

**Supervision:** Erich Seifritz, Roland von Känel, Martin Preisig, Caroline L. Vandeleur.

**Writing – original draft:** Ana Buadze, Vladeta Ajdacic-Gross, Caroline L. Vandeleur.

**Writing – review & editing:** Ana Buadze, Vladeta Ajdacic-Gross, Mario Müller, Yanhua Xu, Erich Seifritz, En-Young N. Wagner, Roland von Känel, Jennifer Glaus, Setareh Ranjbar, Enrique Castelao, Marie-Pierre F. Strippoli, Martin Preisig, Caroline L. Vandeleur.

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
