## [Decision Letter · Decision Letter 0]

9 Jan 2026

PONE-D-25-65436Factors associated with the occurrence and persistence of subthreshold and full attention-deficit hyperactivity disorder in women: a population-based epidemiological studyPLOS One

Dear Dr. Ajdacic-Gross,

Thank you for submitting your manuscript to PLOS ONE. After careful consideration, we feel that it has merit but does not fully meet PLOS ONE’s publication criteria as it currently stands. Therefore, we invite you to submit a revised version of the manuscript that addresses the points raised during the review process.

We look forward to receiving your revised manuscript.

Kind regards,

Mu-Hong Chen, M.D., Ph.D.

Academic Editor

PLOS One

**Journal Requirements:**

“The CoLaus|PsyCoLaus study (represented by MP) is supported by research grants from GlaxoSmithKline (www.gsk.com), the Faculty of Biology and Medicine of the University of Lausanne (https://www.unil.ch/fbm/en/home/menuinst/faculte.html), and the Swiss National Science Foundation (www.snf.ch) (grants 3200B0–105993, 3200B0-118308, 33CSCO-122661, 33CS30-139468, 33CS30-148401, 33CS30_177535 and 3247730_204523) and the Swiss Personalized Health Network (sphn.ch) (grant 2018DRI01).”

“None.”

Reviewers' comments:

Reviewer's Responses to Questions

**Comments to the Author**

1. Is the manuscript technically sound, and do the data support the conclusions?

Reviewer #1: Partly

Reviewer #2: No

Reviewer #3: Yes

2. Has the statistical analysis been performed appropriately and rigorously? 

Reviewer #1: I Don't Know

Reviewer #2: No

Reviewer #3: Yes

3. Have the authors made all data underlying the findings in their manuscript fully available?

Reviewer #1: Yes

Reviewer #2: Yes

Reviewer #3: No

4. Is the manuscript presented in an intelligible fashion and written in standard English?

Reviewer #1: Yes

Reviewer #2: No

Reviewer #3: Yes

5. Review Comments to the Author

Reviewer #1: I have read the above manuscript and give hereafter my comments:

This study covers an interesting issue, yet several aspects are successively described with a neither convincing nor very clear frame.

1. The major concern would be the confounding factors including the psychiatry comorbidities as well as medications seemed not take into consideration of the analyses, especially when analyzed the risk factors of persistent ADHD.

Reviewer #2: This study included a broad range of factors, including family psychiatric comorbidity factors, parental bonding, childhood maltreatment, physical health indicators into analysis to differentiate women with subthreshold and full ADHD. As the authors described, some of these factors were defined as the risk factors and some as “associated” factors. I did not see the hypothesis accounting for the analysis. Further, these were a small number of women having subthreshold or full ADHD. This is far too different from the real-world situation.

Reviewer #3: This study investigated risk factors associated with different subgroups of ADHD in women ADHD, using data from a large Swiss population-based epidemiological cohort. The authors examined a broad range of psychosocial (e.g. adverse childhood experiences, parental bonding), as well as biological and physiological indicators (e.g. childhood infections, cardiometabolic and immunological markers). Overall, this is a data-rich study that addresses an important gap in the ADHD literature, particularly with respect to women and ADHD heterogeneity. The strengths of this manuscript are the large, well-characterized population-based sample, its good representativeness and the richness and breadth of indicators considered. Although some indices are necessarily limited to early-life experiences, the overall multidimensional coverage is impressive. The use of latent class analysis to identify subgroups within subthreshold and full ADHD is a notable contribution, highlighting the substantial heterogeneity of ADHD presentations among women. Overall, I consider this manuscript to be important and novel. My comments below are minor and intended to improve clarity and interpretability.

My primary concern is that the study is highly exploratory and data-driven. This is understandable given the analyses rely on a large, pre-existing epidemiological cohort with a wide array of available measures. However, the manuscript would benefit from a more explicit integrative model of heuristic framework. In particular, the authors could strengthen the discussion by addressing how their findings might translate into practical implication, such as early identification of women at higher risk for persistent ADHD, stratification of ADHD subtypes for prognosis, or potential targets for early intervention.

Minor comments

1. The selection of markers for LCA/LPA was based on a significance threshold p<0.1 for associations with ADHD. The rationale for this decision is unclear and should be more explicitly justified.

2. Table 2. It is unclear why for the same indicator, the total n differs between the subthreshold ADHD group and the full group.

3. When reporting statistics, it would be preferable to report the relevant test statistics (e.g. t value and degrees of freedom), not only the p value (e.g. Line 448), to improve transparency.

4. Line 515 Please check whether the “ACE high BP” group corresponds to Class 3

5. Line 762 The discussion that “lower blood pressure may reflect a distinct subtype of subthreshold ADHD” is intriguing. The authors might consider whether this pattern could relate to concepts such as Cognitive Disengagement Syndrome (or sluggish Cognitive Tempo), autonomic hypoarousal, or alternative neurobiological profiles.

6. PLOS authors have the option to publish the peer review history of their article (what does this mean?). If published, this will include your full peer review and any attached files.

Reviewer #1: No

Reviewer #2: No

Reviewer #3: No

---

## [Author Response · Author response to Decision Letter 1]

24 Feb 2026

The response to Reviewers is uploaded as a file together with other files.

---

## [Decision Letter · Decision Letter 1]

8 Mar 2026

PONE-D-25-65436R1Factors associated with the occurrence and persistence of subthreshold and full attention-deficit hyperactivity disorder in women: a population-based epidemiological studyPLOS One

Dear Dr. Ajdacic-Gross,

Thank you for submitting your manuscript to PLOS ONE. After careful consideration, we feel that it has merit but does not fully meet PLOS ONE’s publication criteria as it currently stands. Therefore, we invite you to submit a revised version of the manuscript that addresses the points raised during the review process.

We look forward to receiving your revised manuscript.

Kind regards,

Mu-Hong Chen, M.D., Ph.D.

Academic Editor

PLOS One

**Journal Requirements:**

**Additional Editor Comments:**

Please optimal answer the reviewer's comment:

The major concern would be the confounding factors including the psychiatry comorbidities as well as medications seemed not take into consideration of the analyses, especially when analyzed the risk factors of persistent ADHD.The authors partially replied to my comment about the confounding factors, including comorbidities and medications. They plan to conduct further analysis about the comorbidities in the future. The authors also argued that medication plays a comparatively minor role in this cohort because pharmacological treatment for ADHD was not available to the majority of participants during their developmental years, young and middle adulthood (median birth year 1952/1953). However, data shows that the FDA has approved Methylphenidate (MPH) for ADHD treatment as early as early 1960s. Therefore the concern about confounding factors still existed.

Reviewers' comments:

Reviewer's Responses to Questions

**Comments to the Author**

1. If the authors have adequately addressed your comments raised in a previous round of review and you feel that this manuscript is now acceptable for publication, you may indicate that here to bypass the “Comments to the Author” section, enter your conflict of interest statement in the “Confidential to Editor” section, and submit your "Accept" recommendation.

Reviewer #1: (No Response)

Reviewer #2: (No Response)

Reviewer #3: All comments have been addressed

2. Is the manuscript technically sound, and do the data support the conclusions?

Reviewer #1: Partly

Reviewer #2: (No Response)

Reviewer #3: (No Response)

3. Has the statistical analysis been performed appropriately and rigorously? 

Reviewer #1: No

Reviewer #2: (No Response)

Reviewer #3: (No Response)

4. Have the authors made all data underlying the findings in their manuscript fully available?

Reviewer #1: No

Reviewer #2: (No Response)

Reviewer #3: (No Response)

5. Is the manuscript presented in an intelligible fashion and written in standard English?

Reviewer #1: Yes

Reviewer #2: (No Response)

Reviewer #3: (No Response)

6. Review Comments to the Author

Reviewer #1: The authors partially replied to my comment about the confounding factors, including comorbidities and medications. They plan to conduct further analysis about the comorbidities in the future. The authors also argued that medication plays a comparatively minor role in this cohort because pharmacological treatment for ADHD was not available to the majority of participants during their developmental years, young and middle adulthood (median birth year 1952/1953). However, data shows that the FDA has approved Methylphenidate (MPH) for ADHD treatment as early as early 1960s. Therefore the concern about confounding factors still existed.

Reviewer #2: (No Response)

Reviewer #3: (No Response)

7. PLOS authors have the option to publish the peer review history of their article (what does this mean?). If published, this will include your full peer review and any attached files.

Reviewer #1: No

Reviewer #2: **Yes:** Cheng-Fang Yen

Reviewer #3: No

---

## [Author Response · Author response to Decision Letter 2]

21 Apr 2026

PONE-D-25-65436R1

Factors associated with the occurrence and persistence of subthreshold and full attention-deficit hyperactivity disorder in women: a population-based epidemiological study

PLOS One

Dear Dr. Chen

Dear Reviewer

Thank you for your follow-up regarding confounding factors in our manuscript. We appreciate the opportunity to continue this discussion and to provide additional context on ADHD in Switzerland.

Please find below our responses to your suggestions and comments. Responses are highlighted in blue.

Kind regards - Vladeta Ajdacic-Gross

Additional Editor Comments / Reviewer 1:

The present comments refer to Comment 1 of the Reviewer on the first version of the manuscript:

"The major concern would be the confounding factors including the psychiatry comorbidities as well as medications seemed not take into consideration of the analyses, especially when analyzed the risk factors of persistent ADHD."

This concern is reflected in the present comment 2 of the Reviewer. Below, we address Part 1:

"The authors partially replied to my comment about the confounding factors, including comorbidities and medications. They plan to conduct further analysis about the comorbidities in the future."

We appreciate the reviewer’s understanding of our research and publication strategy. With regard to psychiatric comorbidities, we acknowledge that these were not systematically included in the present analyses. As noted in the first revision, we have explicitly addressed this as a limitation in the revised manuscript (Methods and Discussion sections) and have outlined plans for future analyses focusing on comorbidity profiles. To date, our analyses have focused on comorbidities involving infectious and other somatic diseases, which typically occur or begin early in life and may act as potential risk factors.

It is important to note that the analysis and interpretation of comorbidities represent one of the most challenging aspects of research on neurodevelopmental and mental disorders, despite appearing straightforward at first glance. These conditions are inherently complex and heterogeneous, and ADHD is no exception.

Analyzing comorbidities involves, among other factors, the intersection of multiple layers of heterogeneity. Consequently, preliminary methodological work is required, for example through network analysis or – as pursued in the present study – through a combined comparative and pattern recognition approach.

The following addresses Part 2 of the reviewer’s comment:

"The authors also argued that medication plays a comparatively minor role in this cohort because pharmacological treatment for ADHD was not available to the majority of participants during their developmental years, young and middle adulthood (median birth year 1952/1953). However, data shows that the FDA has approved Methylphenidate (MPH) for ADHD treatment as early as early 1960s. Therefore the concern about confounding factors still existed."

We agree that the availability of methylphenidate should be considered when evaluating potential confounding by medication. This issue can be viewed from both an empirical and a historical perspective.

Empirical perspective:

The CoLaus|PsyCoLaus cohort study dataset includes information on the use of several medications that may be prescribed for ADHD, as well as for other disorders, including methylphenidate (Ritalin), bupropion (Wellbutrin), lisdexamfetamine (Elvanse), and atomoxetine (Strattera), as well as a category labeled “other stimulants.” Among the 11 women in the CoLaus|PsyCoLaus cohort who reported, either at baseline or during follow-up assessments, having used any of these medications, one woman met criteria for full ADHD, and none met criteria for subthreshold ADHD. As these data refer to lifetime medication use, they also capture aspects of the historical context of treatment exposure.

Historical perspective:

Methylphenidate was introduced in Europe in the 1950s and was initially used for a broad range of non-specific indications (e.g., fatigue, behavioral difficulties), in some contexts even with limited regulatory restrictions. At that time, however, no standardized diagnostic framework for ADHD existed. Early classifications such as “hyperkinetic reaction of childhood” (DSM-II, 1968) and “hyperkinetic syndrome of childhood” (ICD-9, 1975) were narrowly defined, inconsistently applied, and restricted to pediatric populations.

Importantly, ADHD was not recognized as a disorder persisting into adulthood until much later, with broader clinical acceptance emerging only from the 1990s onwards (DSM-IV/ICD-10 era). Consequently, systematic diagnosis – and therefore treatment – during adolescence and adulthood was largely unavailable for this cohort (median birth year 1952/1953).

In Switzerland, the closest related construct at the time was the so-called psycho-organic syndrome (POS), introduced in 1971 as part of the disability insurance framework. Importantly, POS was not a standardized clinical diagnostic category comparable to modern ICD/DSM definitions, but rather an administrative construct with restrictive criteria, and was not broadly implemented in routine psychiatric care.

Taken together, these factors indicate that although methylphenidate was pharmacologically available, its use was unlikely to have been systematic, diagnosis-driven, or widely implemented in this population. As pharmacological treatment presupposes a prior diagnosis, overall exposure to stimulant medication in this cohort is expected to have been low and non-systematically distributed.

In summary, we consider it unlikely that medication exposure constitutes a major source of bias in the present analyses. While regulatory approval or early market availability may suggest availability, this does not necessarily translate into systematic, diagnosis-driven clinical use, particularly across different healthcare systems and historical periods. Nevertheless, we acknowledge that residual confounding cannot be entirely excluded, for example due to underreporting, and we have clarified this limitation in the revised manuscript.

We have supplemented the text as follows (line 841):

"Fourth, information related to symptoms, comorbid diseases / disorders and medications used for therapy was obtained retrospectively and may be subject to recall bias."

---

## [Editor Report · Decision Letter 2]

26 Apr 2026

Factors associated with the occurrence and persistence of subthreshold and full attention-deficit hyperactivity disorder in women: a population-based epidemiological study

PONE-D-25-65436R2

Dear Dr. Vladeta Ajdacic-Gross,

We’re pleased to inform you that your manuscript has been judged scientifically suitable for publication and will be formally accepted for publication once it meets all outstanding technical requirements.

Kind regards,

Mu-Hong Chen, M.D., Ph.D.

Academic Editor

PLOS One
---

## [Editor Report · Acceptance letter]

PONE-D-25-65436R2

PLOS One

Dear Dr. Ajdacic-Gross,

I'm pleased to inform you that your manuscript has been deemed suitable for publication in PLOS One. Congratulations! Your manuscript is now being handed over to our production team.

Kind regards,

on behalf of

Dr. Mu-Hong Chen

Academic Editor

PLOS One